# SHAP-CAT: A INTERPRETABLE MULTIMODAL FRAMEWORK ENHANCING WSI CLASSIFICATION VIA VIRTUAL STAINING AND SHAPLEY-VALUE-BASED MULTIMODAL FUSION

## ABSTRACT

The multimodal model has demonstrated promise in histopathology. However, most multimodal models are based on H&E and genomics, adopting increasingly complex yet black-box designs. Our paper proposes a novel interpretable multimodal framework named SHAP-CAT, which uses a Shapley-value-based dimension reduction technique for effective multimodal fusion. Starting with two paired modalities – H&E and IHC images, we employ virtual staining techniques to enhance limited input data by generating a new clinical-related modality. Lightweight bag-level representations are extracted from image modalities, and a Shapley-value-based mechanism is used to reduce dimensions. For each dimension of the bag-level representation, attribution values are calculated to indicate how changes in the specific dimensions of the input affect the model output. This way, we select a few top critical dimensions of bag-level representation for each imaging modality to late fusion. Our experimental results demonstrate that the proposed SHAP-CAT framework incorporating synthetic modalities significantly enhances model performance, yielding a 5% increase in accuracy for the BCI, an 8% increase for IHC4BC-ER, and an 11% increase for the IHC4BC-PR dataset.

## 1 INTRODUCTION

Recent advances in artificial intelligence have significantly impacted histopathology, mainly by developing multimodal models. These models integrate data types, such as whole slide images and molecular profiles, to improve diagnosis, prediction, and treatment personalization (Chen et al., 2022; Boehm et al., 2022; Chen et al., 2020). Recent efforts are expanding to include multi-staining images like IHC (Jaume et al., 2024; Wang et al., 2024; Foersch et al., 2023) and Trichrome-stained WSIs (Dwivedi et al., 2022) for better identification of specific molecular features related to cancer. Integrating diverse modalities is crucial, since different image modalities carry other information related to cancer (Perez-Lopez et al., 2024; Boehm et al., 2022; Stahlschmidt et al., 2022).

However, many technical, analytical, and clinical challenges are still amplified in the presence of multimodal data.

- Limited public paired datasets (Steyaert et al., 2023; Miotto et al., 2018; Perez-Lopez et al., 2024): Developing multimodal models require modality-paired and datasets with labels. The data also needs to be complete and large in the sample.

- Most multimodal histopathological models combine molecular features and WSIs, not different WSIs. Although molecular data are relevant to precision medicine, they don't have tissue structure, spatial, and morphological information (Alturkistani et al., 2016).

- Very complex and different multimodal fusion technique with *low interpretability*: Li et al. (2022); Wang et al. (2021); Lipkova et al. (2022) have complex design such as hierarchy fusion, intermediate gradual fusion, and intermediate guided-fusion. Still, they ignore the fact that the medical imaging domain requires models to be interpretable.

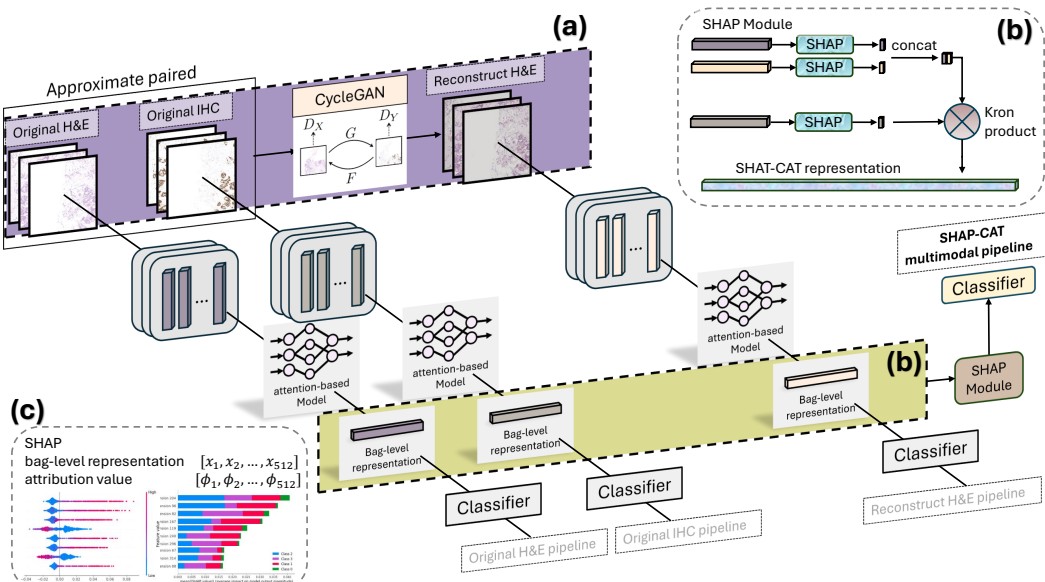

Figure 1: The proposed **SHAP-CAT framework**, which includes three Parallel Feature Extraction Pipelines for different modalities and a SHAP-CAT pipeline for multimodal representation predictions. (a) Generating a new modality by a pre-trained CycleGAN. (b) Extract bag-level representations for each modality from the Parallel Feature Extraction Pipeline and adopt the SHAP pool to reduce dimensions for further late fusion. (c) The key idea is to select the top important dimensions for reduction. The x-axis represents the attribution value, the y-axis ranks features by the magnitude of absolute attributions, and the color indicates the feature value. It's important to note that the meaning of feature values is black-box and hard to interpret. The impact of features can be understood by applying attribution values, and both positive and negative attribution values contribute to the output. (c) left shows the SHAP values of each dimension across all samples within a single class, while the right side shows the mean absolute value of the SHAP values for each dimension, broken down by class in multi-class tasks.

Given the difficulty of obtaining quality datasets (*the first challenge*), we propose a virtual staining-based multimodal framework that uses H&E, IHC, and one more generated modality for WSI classifications. Our multimodal network can integrate triple image modalities in weakly supervised learning on cancer grading tasks (*the second challenge*). After training the specific pipeline and extracting the bag-level representations for each modality, our framework uses the Shapley-value-based dimension reduction approach for further multimodal fusion, avoiding the curse of dimensions and demonstrating high interpretability (*the third challenge*). For a given set of bag-level representations belonging to a patient sample, We employ a Shapley-value-based method to characterize the importance of each dimension within the feature space. This method attributes the predictions of deep neural networks to their respective inputs by computing attribution values for each dimension. This way, we select the top 32 important dimensions for each medical image modality for late fusion and the final classifier for prediction. We evaluate our framework in BCI, IHC4BC-ER, and IHC4BC-PR datasets for cancer grading tasks.

Our contribution is the following:

- A framework with a virtual staining technique is designed to generate one more modality to enhance the limited, approximately paired input dataset without requiring pixel-level data alignment.

- We use a Shapley-value-based mechanism to reduce the dimensions of representation for enhanced multimodal fusion, thereby avoiding the curse of dimensionality and improving the interpretability of our multimodal technique.

- The experiment demonstrates that using virtual staining to generate an additional modality, combined with a Shapley-value-based dimension reduction technique, improves model performance. Specifically, it results in a 5% increase in accuracy for BCI, an 8% increase for IHC4BC-ER, and an 11% increase for IHC4BC-PR.

## 2 RELATED WORK

**Previous general believes on H&E and IHC dataset.** Previous research primarily focuses on image translation and WSI registration algorithms, emphasizing the importance of precise pixel-level alignment for paired medical images (Liu et al., 2022). Competitions like ACROBAT (Weitz et al., 2023) have been organized to advance these technologies, particularly aligning H&E WSIs with IHC WSIs from identical tumor samples. Other studies (Naik et al., 2020; Anand et al., 2021; Shovon et al., 2022) suggest bypassing hard-to-obtain IHC images and predicting cancer and molecular biomarkers using only H&E whole slide images due to accessibility issues.

**Virtual staining technique in medical images.** The deep learning-based virtual staining technique has emerged as an exciting new field that provides more cost-effective, rapid, and sustainable solutions to histopathological tasks. However, this field has no superior measurement standard currently (Latonen et al., 2024). Many studies (Ozyoruk et al., 2022; Levy et al., 2021) rely on pathologists to manually assess the quality of virtually stained images. Others evaluate generated images using traditional metrics like PSNR, SSIM, and FID (de Haan et al., 2021; Vasiljević et al., 2022).

**Multimodal fusion in histopathology.** Several studies (Chen et al., 2020; Li et al., 2022; Wang et al., 2021; Chen et al., 2022) have utilized multimodal techniques to combine histology and genomic data. More and more work designing a very complex multimodal fusion framework. (Wang et al., 2021; Huang et al., 2020; Lipkova et al., 2022; Stahlschmidt et al., 2022). However, there is a lack of research on using common stains like H&E and IHC in multimodal approaches.

## 3 FRAMEWORK DESIGN

The proposed framework consists of Parallel Feature Extraction Pipelines for each modality and a SHAP-CAT pipeline for the predictions of multimodal representations, as illustrated in Fig. 1. Given approximate H&E-IHC paired dataset $I_{he}, I_{ihc}$, we firstly use pre-trained CycleGAN to generate reconstructed H&E images $I_{rec\_he}$. Then we separately train each modality to extract bag-level representations for each modality for further late fusion.

**Modality Generation.** Given the paucity of medical data in general (Zitnik et al., 2019; Miotto et al., 2018), the use of synthetic data has become increasingly prevalent for the training, development, and augmentation of artificial intelligence models (Latonen et al., 2024). We first use the virtual staining technique to generate another modality image for enhancing multimodal framework performance from H&E and IHC paired images, denoted as reconstructed H&E.

The virtual staining technique we used in our paper is CycleGAN (Zhu et al., 2017), designed explicitly for unpaired datasets. The input of our framework is H&E-IHC approximate paired datasets $I_{he}, I_{ihc}$ with labels. Approximate paired here means these two sets of images are not aligned pixel to pixel. In contrast, the same images are offset by about 10%. There are two translators $G : I_{he} \rightarrow I_{ihc}$, and $F : I_{ihc} \rightarrow I_{he}$ (as shown in Fig 1.a). $G$ and $F$ are trained simultaneously to encourages $F(G(I_{he})) \approx I_{he}$ and $G(F(I_{ihc})) \approx I_{ihc}$. Also, there are two adversarial discriminators $D_{he}$ and $D_{ihc}$, where $D_{he}$ aims to discriminate between images $I_{he}$ and translated images $F(I_{ihc})$. Similarly, $D_{ihc}$ aims to distinguish between $I_{ihc}$ and $G(I_{he})$. The final objective is:

$$G^*, F^* = \arg \min_{G,F} \max_{D_{he}, D_{ihc}} \mathcal{L}(G, F, D_{he}, D_{ihc}). \tag{1}$$

The new modality, reconstructed from real H&E-IHC approximate paired images, forms a clinically and biologically relevant pair. Both IHC and reconstructed H&E offer different perspectives of the original H&E slide.

---

**Algorithm 1** The framework of **SHAP-CAT**

---

**Start with**: Approximate paired H&E-IHC staining image $I_{he}, I_{ihc}$ with labels $y$

---

 1: Pre-train a CycleGAN by approximate paired H&E and IHC datasets;
 2: Reconstruct $\{I_{rec\_he}\}_{n=1}^{N_{all}}$ from $\{I_{he}, I_{ihc}\}_{n=1}^{N_{all}}$ by pre-trained CycleGAN;
 3: Preprocess the WSIs $\{I_{he}, I_{ihc}, I_{rec\_he}\}_{n=1}^{N_{all}}$ and extract features $\{R_{he}, R_{ihc}, R_{rec\_he}\}_{n=1}^{N_{all}}$;
 4: Data splitting of $\{D\}_{n=1}^{N_{all}} \rightarrow \{D_1\}_{n=1}^{N_{train}}, \{D_2\}_{n=1}^{N_{val}}, \{D_3\}_{n=1}^{N_{test}}$;
 5: **while  (Parallel Feature Extraction Pipeline)  do**
 6:     **for** each modality **do**
 7:        $Model.fit(R, y)$ on $\{D_1\}_{n=1}^{N_{train}}$ with $\{D_2\}_{n=1}^{N_{val}}$;
 8:        $\hat{y} \leftarrow Model(R)$ on $\{D_3\}_{n=1}^{N_{test}}$ to obtain the performance for single modality pipeline;
 9:        extract bag-level representation $z$ at the penultimate hidden layer;
10:     **end for**
11: **end while**
12: **while (SHAP-CAT multimodal pipeline) do**
13:     Apply SHAP pooling $\sigma$ to reduce dimensions for $z_{he}, z_{ihc}, z_{rec\_he}$, respectively;
      $f_{he} \leftarrow \sigma_{he}(z_{he})$, $f_{ihc} \leftarrow \sigma_{ihc}(z_{ihc})$, $f_{rec\_he} \leftarrow \sigma_{rec\_he}(z_{rec\_he})$, where $f \in \mathbb{R}^{1 \times 32}$;
14:     Concat two H&E representations: $f_{he\_final} \leftarrow [f_{he}, f_{rec\_he}]$, where $f_{he\_final} \in \mathbb{R}^{1 \times 64}$;
15:     Fusion three modalities: $F = f_{he\_final} \otimes f_{ihc}$, where $F \in \mathbb{R}^{1 \times 2048}$;
16: **end while**
17: Mapping of $F \rightarrow y$;
    $y \leftarrow classifier(F)$ on $\{D_1\}_{n=1}^{N_{train}}$;
18: Obtaining the performance for SHAP-CAT multimodality pipeline:
    $\hat{y} \leftarrow classifier(F)$ on $\{D_3\}_{n=1}^{N_{test}}$.

---

**Parallel Feature Extraction Pipeline.**    In this paper, the three modalities used—H&E, IHC, and reconstructed H&E images—are each assigned to a specific feature extraction pipeline. For each input WSI denoted as "bag" in the standard attention-based MIL pipeline (Ilse et al., 2018; Lu et al., 2021), the bag $I$ is split into $K$ patches $I = \{I(1), I(2), \cdots, I(K)\}$, where $x$ is denoted as "instance" and $K$ varies for different input. Each bag will be pre-processed and then extract feature $R = \{r_1, r_2, \cdots, r_K\}$. There are N such bag with their label $y$ constituting the dataset $\{D\}_{n=1}^{N_{all}}$. During training, the whole dataset will be split into training $\{D_1\}_{n=1}^{N_{train}}$, validating $\{D_2\}_{n=1}^{N_{val}}$, and testing $\{D_3\}_{n=1}^{N_{test}}$ subset, where $\{I_{he}, I_{ihc}, I_{rec\_he}\}_{n=1}^{N_{all}}$ sharing the same data splitting subset.

The embedding $r_k$ is compressed by a fully connected layer to $\boldsymbol{h}_k$. Then $h_k$ is fed into the multi-class classification network, aggregating the set of embeddings $h_k$ into a bag-level embedding $\boldsymbol{z}_n = \sum_{k=1}^{K} a_{k,n} \boldsymbol{h}_k$, where Eq. 2 computes the attention scores for the $k-thinstancefequation: gated-attention-here$.

$$a_{k,n} = \frac{\exp\{W_{atten,n}(\tanh(Vh_k^T) \odot sigm(Uh_k^T))\}}{\sum_{j=1}^{K} \exp\{W_{atten,n}(\tanh(Vh_j^T) \odot sigm(Uh_j^T))\}} \tag{2}$$

Finally, the bag-level representation $z_n$ is extracted at the penultimate hidden layer before the last classifier.

**SHAP-CAT Fusion Module.**    Once the bag-level representations have been constructed from each modality, a SHAP-CAT fusion module is introduced to capture informative inter-modality interactions between H&E, IHC, and reconstructed H&E features. Before late fusion, we propose an efficient and highly interpretable SHAP pool to reduce dimensions of bag-level representations $z$ to avoid the curve of dimensions. We model the dimension reduction as an attribution problem that attributes the prediction of machine learning models to their inputs (Lundberg & Lee, 2017; Ribeiro et al., 2016; Shrikumar et al., 2017). For bag-level representations $z = [d_1, d_2, \ldots, d_{512}] \in \mathbb{R}^{1 \times 512}$, each dimension $d$ has attribution values corresponding to the contributions toward the model prediction. Dimensions that have no effect on the output are assigned zero attribution, suggesting no relevance, whereas dimensions that significantly influence the output exhibit higher attribution val-

ues, indicating their importance. As illustrated in Fig 1(c), we visualize the attribution values of each dimension to understand the magnitude of how much it impacts the output.

The proposed SHAP pool selects each modality's top 32 essential dimensions and then applies the Kronecker product as late fusion. This module constructs the joint representations as the input of the final prediction for multimodalities. The whole algorithm is shown in Algorithm 1. We further introduce Shapley-value-based dimension reduction and multimodal fusion in the next section.

# 4 EXPLAINABLE MULTI-MODAL FUSION

In this section, we define the impact of dimension reduction in multimodal technique as an attribution problem, quantifying how the changes of dimensions within input representations affect the model output.

## 4.1 PROBLEM FORMULATION

Given a set of inputs $\{z_n\}_{n=1}^N$ where $z = [d_1, d_2, \ldots, d_{512}] \in \mathbb{R}^{1 \times 512}$ and a model $f(z)$, the output changes when dimensions within $z$ vary. Each dimension $d_i$ can interact with each other. Therefore, we define the attribution problem as follows: each dimension $d_i$ has its attribution value $\phi_i$, which indicates how much it impacts the output. The goal is to determine the attribution values $\{\phi_1, \phi_2, \cdots, \phi_{512}\}$ of input bag-level representations $\{z_n\}_{n=1}^N$ by computing the contribution of each dimension within $z$ to the model output. We simplify the problem into:

$$\{z_n\}_{n=1}^N = \begin{bmatrix} d_{1,1} & d_{1,2} & \cdots & d_{1,512} \\ \vdots & \vdots & \vdots & \vdots \\ d_{N,1} & d_{N,2} & \cdots & d_{N,512} \end{bmatrix} = \{x_1, x_2, \cdots, x_{512}\} \Rightarrow \{\phi_1, \phi_2, \cdots, \phi_{512}\} \quad (3)$$

In our paper, Shapley value (Shapley et al., 1953), a game theory solution to denote a player's marginal contribution to the payoff of a coalition game, is employed to measure the impact of individual dimension within representations for a model.

There is a characteristic function $v$ that maps subsets $S \subseteq \{x_1, x_2, \cdots, x_{512}\}$ to a real value $v(S)$, which represents how much payoff a set of dimensions can gain by "comperating" as a set. $v(S)$ measures the importance of dimensions by sets. Now, we move on to the single dimension. The marginal contribution $\triangle_v(i, S)$ of the specific dimension features $x_i$ with respect to a subset $S$ is denoted as $\triangle_v(i, S) = v_{S \bigcup\{i\}}(x_{S \bigcup\{i\}}) - v_S(x_S)$. Intuitively, the Shapley value can be defined as the weighted average of the specific dimension's marginal contributions to all possible subsets of dimensions.

**Definition 1** *Shapley values* *quantifies the importance of each useful dimension by marginal contribution*

$$\phi_i = \sum_{S \subseteq N \setminus \{i\}} \frac{\mid S \mid! (\mid N \mid - \mid S \mid - 1)!}{\mid N \mid!} [v_{S \bigcup\{i\}}(x_{S \bigcup\{i\}}) - v_S(x_S)],$$

The above formula is a summation over all possible subsets $S$ of feature values excluding the $x_i$'s value. $\phi_i$ is a unique allocation of the coalition and can be viewed as the influence of $x_i$ on the outcome. Therefore, the question becomes – how to identify $\{\phi_1, \phi_2, \cdots, \phi_{512}\}$ of bag-level representations for a machine learning model.

## 4.2 INTERPRETABILITY IN MACHINE LEARNING

To obtain attribution values for each dimension, we must first explain the machine learning model. For complex models in machine learning, its explanation can be represented by a simpler explanation model (Ribeiro et al., 2016; Lundberg & Lee, 2017; Shrikumar et al., 2017). The simplified explanatory model is defined as an interpretable approximation of the original model. The original model that needs to be explained is given as $f$. $g$ is the explanation model to explain $f$ based on

the single dimension $x_i$ of feature: $f(x) = g(x')$. Explanation models distinguish an interpretable representation from the original feature space that the model uses. The function $x = h_x(x')$ is applied to map the original value $x$ to a simplified input $x'$, where $x' \in \{0,1\}^M$, M is the max number of coalition, and $\phi_i \in R$. The simplified input $x'$ maps 0 or 1 to the corresponding feature value, indicating the present or absent state of the corresponding feature value.

**Definition 2** *Mapping feature value into simplified input*

$$x' = \left\{ (x^p)' = 1, \quad (x^a)' = 0 \begin{cases} x_i = 0 \\ x_i \neq 0, \ but \ v_{S\bigcup\{i\}}(x_{S\bigcup\{i\}}) = v_S(x_S) \forall S \end{cases} \right\}$$

where $x^p$ means the presence of a feature and $x^a$ means the absence of a feature; we will discuss them in Section 4.3. $\phi_i$ is the attribution value of $x_i$, corresponding the the specific dimension $d_i$ for bag-level representations. The function $x = h_x(x')$ maps 1 to the specific dimension that we want to explain and maps 0 to the values of the specific dimension that has no attributed impact on the model.

**Property 1** *Meaningless dimension*

$$x_i' = 0 \Rightarrow \phi_i = 0$$

After turning a feature vector into a discrete binary vector, we can define the attribution values for the model. For an explanatory model to have additive feature attribution, the explanatory model could be expressed as the sum of the null output of the model and the summation of explained effect attribution.

**Property 2** *Local accuracy*

$$f(x) = g(x') = \phi_0 + \sum_{i=1}^{M} \phi_i x_i',$$

Explanation models also exhibit a property known as consistency, stating that if a model changes and makes a contribution of a particular feature stays the same or increases regardless of other inputs, the attribution assigned to that feature should not decrease.

**Property 3** *Consistency*

$$v_1(S\bigcup\{i\}) - v_1(S) \geq v_2(S\bigcup\{i\}) - v_2(S) \forall S \Rightarrow \phi_i(v_1, x) \geq \phi_i(v_2, x)$$

Combining the information from Sections 4.1 and 4.2, we can find that the Shapley value is the only solution to satisfy the three properties of the explanatory models. Now, we get the explanation models related to attribution values $\phi_i$. The new question is – how to estimate it?

## 4.3 SHAPLEY VALUE OF FEATURE DIMENSION

From all the previous property and definition, we have $\phi_0 = E[f(x)] = f(\emptyset)$. So the Property 2 will be $f(x) = g(x') = \sum_{i=1}^{M} \phi_i x_i'$, stating that when approximating the model $f$ for input $x$, the explanation's attribution values $\phi_i$ for each feature $x_i$ should sum up to the output $f(x)$. We aim to obtain local feature attributions $\phi_i$, a vector of importance values for each feature of a model prediction for a specific sample $x_i$.

According to Definition 2, if feature $x_i$ is present, we can simply set that feature to its value in $x^p$. The next step is to address the absence of a feature $x^a$.

One approach to incorporate $x^a$ into the coalitional game is with a conditional expectation. We condition the set of features that are "present" as if we know them and use those to guess at the "missing" features, so the value of the game is: $v(S) = E_D[f(x)|x_S]$. Therefore, $\phi_i(f, x^p) = \frac{1}{|D|} \sum_{x^a \in D} \phi_i(f, x^p, x^a)$, where $D$ is the distribution of $x^a$. In summary, obtaining $\phi_i(f, x^p)$ reduces to an average of simpler problems $\phi_i(f, x^p, x^a)$, where our $x^p$ is compared to a distribution with only one sample $x^a$.

In our paper, we employ treeSHAP (Lundberg et al., 2020), designed as a fast alternative for tree-based ML models such as random forests or decision trees, to calculate $\phi_i(f, x^p, x^a)$. Computational complexity is reduced to $O(TLD^2)$ where $D$ is the maximum depth of any tree, $L$ is the number of leaves and $T$ is the number of trees.

Given bag-level representation $\{z_n\}_{n=1}^{N} \in \mathbb{R}^{N \times 512}$ with labels $y$, we train a random forest classifier on $\{z_n, y_n\}_{n=1}^{N}$ for estimation to obtain the attribution value $[\phi_1, \phi_2, \ldots, \phi_{512}]$ for dimension reduction. The whole SHAP pool is demonstrated in Algorithm 2.

---

**Algorithm 2 (SHAP pool)**

---

**Input**: $\{z\}_{n=1}^{N_{all}}$ with label $\{y\}_{n=1}^{N_{all}}$, where $z = [d_1, d_2, \ldots, d_{512}] \in \mathbb{R}^{1 \times 512}$
**Output**: $\{f\}_{n=1}^{N_{all}}$, where $f \in \mathbb{R}^{1 \times 32}$
1: $z_{train}, z_{test}, y_{train}, y_{test} \leftarrow Data\_Split(z, y)$;
2: model = RandomForestClassifier( );
3: model.fit($z_{train}, y_{train}$);
4: shap_values $\leftarrow$ treeSHAP $(model, z_{test})$;
   $[\phi_1, \phi_2, \ldots, \phi_{512}] \leftarrow [x_1, x_2, \ldots, x_{512}]$, where $\phi$ is the attribution value of each dimension;
5: select top 32 shap_values $\phi_i$ for $z$;
6: Dimension reduction:
   $f \leftarrow \sigma(z)$, where $z \in \mathbb{R}^{1 \times 512}$ and $f \in \mathbb{R}^{1 \times 32}$.

---

### 4.4 FUSION OF MODALITY

In multimodal fusion, direct fusion of multiple modalities is impractical. For example, bag-level representation of each modality is represented in 512 dimensions in our paper. Consequently, three dimensions would generate features of $512^3$ dimensions, making it impractical for machine learning model training. In addition, such large-dimension data face a challenge known as the curse of dimensionality. Furthermore, trying to tackle complex histopathological tasks with such high-dimensional yet low-sample-size features results in "blind spot" (Berisha et al., 2021).

Therefore, we must decrease the dimensionality of representations. Prior research has utilized average pooling or max pooling for this purpose (Wang et al., 2024; Chen et al., 2020). Our method deviates from traditional methods by offering a more accurate and interpretable strategy for fusion. We are the first to implement a Shapley-value-based technique to reduce dimensions in image modality representations. We also evaluate our SHAP pool in a single modality by reducing bag-level representation $z \in \mathbb{R}^{1 \times 512}$ to $f \in \mathbb{R}^{1 \times 32}$ and then aggregated by different classifiers (as shown in Tab 1). We compare our SHAP pooling with average pooling, max pooling and selecting 32 dimensions randomly. Our SHAP pooling performs well across different classifiers.

**Generate low-dimension features by SHAP Pooling.** From the Parallel Feature Extraction Pipeline, we extract bag-level feature representations $z_{he}$, $z_{rec\_he}$ and $z_{ihc}$. We adopt the proposed shaply-value-based pooling to fuse H&E, IHC and reconstructed H&E representations. Using SHAP pooling $\sigma$, we select the most important 32 dimensions of original bag-level representation $z_{he}$, $z_{rec\_he}$ and $z_{ihc}$ to generate low-dimension representation $f_{he}$, $f_{rec\_he}$ and $f_{ihc} \in \mathbb{R}^{1 \times 32}$ by $f = \sigma(z)$.

**Kronecker product.** IHC is a staining technique that visualizes the overexpression of target proteins. The visualized locations help understand the morphological characteristics of cells within a tissue. Thus, IHC and H&E WSIs provide different information on molecular features. For the modality from true H&E and reconstructed H&E whole slide images, $z_{he}$ and $z_{rec\_he}$ are directly concatenated to generate the new representation $f_{he\_final} \in \mathbb{R}^{1 \times 64}$ of H&E staining images: $f_{he\_final} = [f_{he}, f_{rec\_he}] = [\sigma(z_{he}), \sigma(z_{rec\_he})]$. In order to capture the intricate relationships between H&E and IHC modalities, we follow previous work (Wang et al., 2021; Chen et al., 2020; Wang et al., 2024; Chen et al., 2022; Li et al., 2022) that employ Kronecker product, denoted as $\otimes$, to fuse different modalities. Therefore, the joint multimodal tensor $F \in \mathbb{R}^{1 \times 2048}$ constructed from the Kronecker product, as shown in Eq.(8), will capture the important interactions that characterize H&E and IHC modalities.

Table 1: Effectiveness of Proposed Shap Pooling.

| Model | Accuracy | | | | | |
|---|---|---|---|---|---|---|
| | **SHAP** | Avg | Max | Rand1 | Rand2 | Rand3 |
| Random Forest | **0.898** | 0.867 | 0.849 | 0.821 | 0.829 | 0.808 |
| SVM | **0.900** | 0.862 | 0.852 | 0.785 | 0.795 | 0.762 |
| Logistic Regression | **0.862** | 0.765 | 0.793 | 0.734 | 0.698 | 0.734 |
| KNeighbors | **0.903** | 0.903 | 0.893 | 0.793 | 0.847 | 0.806 |
| Decision Tree | **0.824** | 0.760 | 0.777 | 0.734 | 0.739 | 0.721 |
| MLP (ours) | **0.885** | 0.821 | 0.859 | 0.777 | 0.806 | 0.767 |
| XGB Classifier | **0.903** | 0.882 | 0.875 | 0.816 | 0.847 | 0.811 |
| LGBM Classifier | **0.900** | 0.885 | 0.880 | 0.818 | 0.849 | 0.813 |
| CatBoost (ours) | **0.903** | 0.875 | 0.880 | 0.839 | 0.847 | 0.844 |

$$F = f_{he\_final,n} \otimes f_{ihc,n} = [\sigma(z_{he}), \sigma(z_{rec\_he})] \otimes \sigma(z_{ihc,n}) \qquad (8)$$

After constructing the joint representation, we use the multimodal representation $F$ as input. It is then processed by classifiers like MLP or CatBoost (Prokhorenkova et al., 2018) for cancer grading tasks.

## 5 EXPERIMENTS

**Datasets and Implementation Details**   We use two public datasets BCI (Liu et al., 2022) and IHC4BC (Akbarnejad et al., 2023) in this paper. Both of them are cancer grading tasks. We use CLAM (Lu et al., 2021) as the pre-processing tool and original training pipeline. The details are shown in the Appendix.

**Results on BCI and IHC4BC datasets**   Tab 2 shows the detailed results on the BCI dataset, and Tab 3 presents the results on the IHC4BC-ER and IHC4BC-PR datasets. Most previous models only deal with a single modality. Multiple modalities achieve higher performance than all models in a single modality. Our SHAP-CAT method includes modality enhancement via virtual staining, efficient multimodal fusion by Shapley-value-based dimension reduction, and finally, aggregation in the MLP or CatBoost classifiers (Prokhorenkova et al., 2018), achieving higher accuracy across BCI and IHC4BC datasets.

Table 2: **Experiment Results on the BCI Dataset.** The performance is reported as AUC and ACC.

| Model | Modality | Performance | |
|---|---|---|---|
| | | **AUC** | **ACC** |
| InceptionV3 (Szegedy et al., 2016) | H&E | 0.823 | 0.804 |
| ResNet (He et al., 2016) | H&E | 0.886 | 0.872 |
| ViT (Ayana et al., 2023) | H&E | 0.92 | 0.904 |
| HAHNet (Wang et al., 2023) | H&E | 0.99 | 0.937 |
| DenseNet (Huang et al., 2017) | H&E | 0.890 | 0.68 |
| HE-HER2Net (Shovon et al., 2022) | H&E | 0.980 | 0.870 |
| ABMIL (Ilse et al., 2018) | H&E | 0.985 | 0.902 |
| ABMIL | IHC | 0.991 | 0.916 |
| CLAM (Lu et al., 2021) | H&E | 0.987 | 0.909 |
| CLAM | IHC | 0.991 | 0.917 |
| TransMIL (Shao et al., 2021) | H&E | 0.991 | 0.907 |
| TransMIL | IHC | 0.994 | 0.931 |
| **Shap-cat Fusion + MLP (ours)** | **H&E, rec H&E, IHC** | **0.997** | **0.959** |
| **Shap-cat Fusion + CatBoost (ours)** | **H&E, rec H&E, IHC** | **0.996** | **0.955** |

**Reconstruct modality enhance the performance for multimodal model**   As mentioned in the previous section, our framework uses the CLAM pipeline to extract the bag-level representations $z$.

Therefore, we also report the performance of the baseline trained by reconstructed H&E modality. As shown in Tab 2 and Tab 3, the reconstructed H&E modality generated by CycleGAN results in lower performance when it is used as the main input for the single-modality model. However, it can enhance multimodal model performance when we use our SHAP-CAT fusion to efficiently capture information across three modalities. In the BCI dataset, three original pipelines, which train H&E, IHC, and rec H&E modalities separately to extract bag-level representations $z_{he}, z_{ihc}, z_{rec\_he}$, achieve accuracy in 0.909, 0.917 and 0.787. However, their multimodal representations can be aggregated by the classifier in much higher results, achieving 0.959 in accuracy. This situation also occurs in IHC4BC datasets.

Table 3: **Experiment Results on IHC4BC Dataset.** The performance is reported as AUC and ACC for IHC4BC-ER and IHC4BC-PR.

| Model | Modality | IHC4BC-ER | | IHC4BC-PR | |
|---|---|---|---|---|---|
| | | **AUC** | **ACC** | **AUC** | **ACC** |
| ABMIL | H&E | 0.953 | 0.843 | 0.911 | 0.835 |
| ABMIL | IHC | 0.978 | 0.888 | 0.959 | 0.841 |
| CLAM | H&E | 0.9543 | 0.8421 | 0.908 | 0.777 |
| CLAM | IHC | 0.979 | 0.894 | 0.957 | 0.84 |
| Transmil | H&E | 0.95 | 0.851 | 0.911 | 0.791 |
| Transmil | IHC | 0.979 | 0.902 | 0.959 | 0.85 |
| **Shap-cat Fusion + MLP (ours)** | **H&E, rec H&E, IHC** | 0.98 | **0.925** | 0.921 | **0.877** |
| **Shap-cat Fusion + CatBoost (ours)** | **H&E, rec H&E, IHC** | **0.985** | **0.928** | **0.969** | **0.883** |

## 6 ABLATION STUDY

In our paper, we use the following strategies:

- Strategy 1 : virtual staining to generate reconstructed H&E
- Strategy 2 : SHAP pooling to reduce the dimension of original bag-level representation

We evaluate our virtual staining strategy. Since we use CLAM to extract bag-level representations, we compare single, double, and triple modalities in Table 4. Also, we compare the results of two modalities(H&E-IHC) with three modalities(H&E, IHC, and reconstructed H&E) processed by the same pooling across different classifiers as aggregations in Table 5. What's more, we evaluate our SHAP pool with average pool across different classifiers in Table 5.

Table 4: **Ablation Study of Virtual Staining on BCI and IHC4BC datasets.** Results are reported as AUC and ACC for each modality.

| Model | Modality | BCI | | IHC4BC-ER | | IHC4BC-PR | |
|---|---|---|---|---|---|---|---|
| | | **AUC** | **ACC** | **AUC** | **ACC** | **AUC** | **ACC** |
| CLAM | H&E | 0.987 | 0.909 | 0.954 | 0.842 | 0.908 | 0.777 |
| CLAM | IHC | 0.991 | 0.917 | 0.979 | 0.894 | 0.957 | 0.84 |
| CLAM | rec H&E | 0.937 | 0.787 | 0.949 | 0.835 | 0.916 | 0.783 |
| Shap-cat + MLP | H&E, IHC | 0.995 | 0.941 | 0.984 | 0.91 | 0.919 | 0.866 |
| Shap-cat + CatBoost | H&E, IHC | 0.994 | 0.946 | 0.985 | 0.911 | 0.967 | 0.875 |
| Shap-cat + MLP | H&E, rec H&E, IHC | 0.997 | 0.959 | 0.98 | **0.925** | 0.921 | **0.877** |
| Shap-cat + CatBoost | H&E, rec H&E, IHC | 0.996 | 0.955 | **0.985** | **0.928** | **0.969** | **0.883** |

## 7 DISCUSSION

### 7.1 VIRTUAL STAINING CAN BE USED FOR ENHANCING NOT MAIN INPUT

Limited labeled datasets are a crucial challenge for the whole histopathology field, especially for multimodal models. Our framework applies a virtual staining technique to enhance WSI classification, providing a different solution. Our synthesis data satisfy the following requirements suggested

Table 5: **Ablation Study of SHAP pooling on BCI dataset**. The results are reported as the average AUC and ACC metrics for two and three multimodal settings.

| Framework | Model | Two Multimodal | | Three Multimodal | |
|---|---|---|---|---|---|
| | | AUC | ACC | AUC | ACC |
| Avg Bilinear Fusion | RandomForest | 0.994 | 0.936 | 0.997 | 0.941 |
| | Logistic Regression | 0.982 | 0.844 | 0.986 | 0.890 |
| | DecisionTree | 0.898 | 0.857 | 0.892 | 0.844 |
| | MLP | 0.989 | 0.923 | 0.995 | 0.944 |
| | GaussianNB | 0.963 | 0.872 | 0.949 | 0.862 |
| | CatBoostClassifier | 0.990 | 0.928 | 0.996 | 0.944 |
| Shap-cat Multimodal Fusion | RandomForest | 0.993 | 0.944 | 0.996 | 0.951 |
| | Logistic Regression | 0.995 | 0.928 | 0.994 | 0.932 |
| | DecisionTree | 0.891 | 0.872 | 0.903 | 0.883 |
| | MLP | 0.995 | 0.941 | 0.997 | **0.959** |
| | GaussianNB | 0.967 | 0.918 | 0.996 | 0.956 |
| | CatBoostClassifier | 0.994 | 0.946 | 0.996 | 0.955 |

by the FDA AI/ML white paper and 21st Century Cures Act (Steyaert et al., 2023): (1) relevant to the clinical practice and clinical endpoint; (2) collected in a manner that is consistent, generalizable, and clinically relevant; and (3) output is appropriately transparent for users.

We claim that virtual staining may not be good for the training model as the main input, but it is good for enhancing performance as an extra modality. As shown in Table 5, our reconstructed modality performs well across different classifiers, compared to the single or double modality.

## 7.2 DENSE BAG-LEVEL REPRESENTATION

SHAP value (Lundberg & Lee, 2017) is an approximation of Shapley values, while the original Shapley value (Shapley et al., 1953) is an NP-hard problem in game theory. It is impossible to search for an NP-hard problem directly in features extracted from giga-pixel WSIs. The bag-level representation generated by our framework is a 512-dimension feature with a size of 3.3kb. There are $1 \times 512$ elements in the tensor. Each element is 4 bytes. Therefore, the total data size is 2048 bytes (or exactly 2 KB for the data). The raw data size is about 2 KB. The actual file size might be slightly larger due to the metadata and can vary slightly based on the specific version of PyTorch and the details of how the tensor storage is implemented. Similarly, our final bag-level representation is a 2048-dimension tensor, which is only 9.4kb in size. This very small size ensures us to use many models to aggreate the final bag-level representation.

## 8 CONCLUSION

We propose a novel framework with a virtual staining technique to generate one more modality to enhance WSI classification and a Shapley-value-based mechanism to reduce dimensions for efficient and interpretable multimodal fusion for histopathological tasks. We are the first to use the Shapley-value-based dimension-reducing technique in image modality. The experiment demonstrates that using virtual staining to generate an additional modality, combined with a Shapley-value-based dimension reduction technique, improves model performance. Specifically, it results in a 5% increase in accuracy for BCI, an 8% increse for IHC4BC-ER and an 11% increase for IHC4BC-PR.

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

# A APPENDIX

## A.1 DATASET

We use two public breast cancer datasets in this paper. BCI dataset (Liu et al., 2022) presents 4870 registered H&E and IHC pairs, covering a variety of HER2 expression levels from 0 to 3. IHC4BC dataset (Akbarnejad et al., 2023) contains H&E and IHC pairs in ER and PR breast cancer assessment, and categories are defined ranges 0 to 3 respectively. The number of each subset is 26135 and 24972.

## A.2 IMPLEMENTATION DETAILS

We use CLAM (Lu et al., 2021) pre-processing tools to create patches and extract features from each WSI image. Some WSIs will be dropped due to the segment and filtering of the CLAM pre-processing mechanism; we take the intersection of H&E and IHC pre-processed WSIs for further training. The learning rate of the Adam optimizer is set to $2 \times 10^{-4}$, the weight decay is set to $1 \times 10^{-5}$, the early-stop strategy is used, and the max training epochs are 200. We trained our multimodal model using a weakly supervised paradigm in 5-fold Monte Carlo cross-validation and

performed ablation analysis to compare the performance between unimodal and multimodal prognostic models. For each cross-validated fold, we randomly split each dataset into 80%-10%-10% subset of training, validation, and testing, stratified by each class.

### A.2.1 THE TRAINING PROCESS FOR A SINGLE BRANCH

Suppose we have a WSI image $I$, which will denoted as "bag" in the following description. The bag $I$ is split into $K$ patches $I = \{I(1), I(2), \cdots, I(K)\}$, which is denoted as "instance". Each instance will be preprocessed and then fed into ResNet50 (He et al., 2016) to get the embedding. Let $R = \{r_1, r_2, \cdots, r_K\}$ be the embedding result from feature extraction for bag $I$ of $K$ instance, $r_k \in \mathbb{R}^{1 \times 1024}$. The first fully-connected layer $W_{fc} \in \mathbb{R}^{512 \times 1024}$ compresses each embedding $r_k$ to $h_k$, a denser feature vector.

$$\boldsymbol{h}_k = W_{fc}\boldsymbol{r}_k^T \in \mathbb{R}^{1 \times 512} \tag{9}$$

The denser feature vector $h_k$ is then fed into the multi-class classification network, which consists of attention module and slide-level classifiers. Given $n$ class, the attention network will be split into $n$ parallel $W_{atten,1}, W_{atten,2}, \cdots, W_{atten,n} \in \mathbb{R}^{1 \times 256}$. For multi-class classification, the attention module in Attention-Based MIL aggregates the set of embeddings $h_k$ into a bag-level embedding $z_n$ by Eq. 10, where the attention for the k-th instance is computed by Eq. 11.

$$\boldsymbol{z}_n = \sum_{k=1}^{K} a_{k,n}\boldsymbol{h}_k \tag{10}$$

$$a_{k,n} = \frac{\exp\left\{W_{atten,n}(\tanh(Vh_k^T) \odot sigm(Uh_k^T))\right\}}{\sum_{j=1}^{K} \exp\left\{W_{atten,n}(\tanh(Vh_j^T) \odot sigm(Uh_j^T))\right\}} \tag{11}$$

where $a_{k,n}$ is the confidence that $k^{th}$ instance belongs to $n^{th}$ class, which denoted as "attention score". $U, V \in \mathbb{R}^{256 \times 512}$ are learnable attention backbone shared for each class in the attention mechanism, and $\odot$ is the element-wise multiplication for the gated attention mechanism. $z_n \in \mathbb{R}^{1 \times 512}$ is the weighted sum of input $h_k$ for the $n^{th}$ class. In this way, the input feature $R = \{r_1, r_2, \cdots, r_K\}$ extracted by ResNet are encoded into a dense feature vector $z_n$ as bag-level representation. We emphasize that the instance number in the bag would not influence the output shape of $z_n$, as each instance embedding is computed with its attention score to generate the bag-level representation for the $n^{th}$ class.

Then, for the original complete training pipeline, the bag-level representation $z_n$ is utilized for predicting the bag-level (also called slide-level) score $s_n$. The bag-level score is computed by a group of classifiers $\{W_{c,1}, \cdot, W_{c,n}\}$ as Eq.(12):

$$s_n = W_{c,n}z_n^T \tag{12}$$

The bag-level score $s_n$ will be further fed into $softmax(s_n)$ for the final prediction.

