# OpenReview forum: "SHAP-CAT: A interpretable multi-modal framework enhancing WSI classification via virtual staining and shapley-value-based multimodal fusion"
_ICLR.cc/2025/Conference — ICLR 2025 Conference Withdrawn Submission_

### Official Review · Reviewer_tgf5 · 2024-10-30

**Soundness:** 1
**Presentation:** 1
**Contribution:** 2
**Rating:** 1
**Confidence:** 4

**Summary:**

This paper proposed a new multimodal framework for WSI classification. The proposed method is validated on two datasets.

**Strengths:**

A new multimodal framework for WSI classification.

**Weaknesses:**

1) The experimental results are not as promising as claimed in the abstract. The improvement in classification performance is minimal.
2The paper does not address the proposed problem as effectively as the authors claim.

**Questions:**

1）How does the proposed method address the issue of a limited dataset?
2)   Many multimodal fusion methods have been proposed, and this paper lacks a comparison with these existing methods.
3）The paper does not present interpretable results.
4）The classification results do not demonstrate the effectiveness of the proposed method.
5）The method description lacks sufficient detail.

---

### Official Review · Reviewer_TbvU · 2024-11-04

**Soundness:** 3
**Presentation:** 3
**Contribution:** 3
**Rating:** 5
**Confidence:** 4

**Summary:**

The paper introduces SHAP-CAT, an innovative and interpretable multimodal framework designed to classify whole-slide images (WSI) in cancer histopathology. SHAP-CAT aims to enhance classification accuracy by integrating multiple image modalities and leveraging interpretability through Shapley values. The framework focuses on Hematoxylin and Eosin (H&E) stained images, ImmunoHistoChemistry (IHC) images, and a synthetically generated modality derived via virtual staining. Experiments shows consistent increase in accuracy for three datasets.

**Strengths:**

The application of CycleGAN for generating a synthetic reconstructed H&E modality is a creative solution to data scarcity, enabling the framework to leverage additional information without the need for expensive and time-consuming data acquisition. By avoiding the necessity for precise pixel-level alignment, the model simplifies preprocessing steps and broadens its applicability across various datasets.

Incorporating SHAP values for dimension reduction not only optimizes the feature space but also maintains interpretability, enabling clinicians to understand which features are most influential in model predictions.

**Weaknesses:**

The quality and reliability of the synthetic reconstructed H&E images depend heavily on the CycleGAN's performance. Poorly generated synthetic images could introduce noise or artifacts, potentially degrading model performance. And there is a lack of quality control on CycleGAN results.

The combination of CycleGAN for virtual staining, Transformer-based feature extraction (if applicable), and Shapley value computations can be computationally demanding, potentially limiting scalability to larger datasets or real-time applications.

Dataset Diversity: The model was evaluated primarily on two breast cancer datasets. Testing on a more diverse set of cancer types and datasets would be better.

Choice of Kronecker Product: The rationale behind selecting the Kronecker product for multimodal fusion could be more thoroughly justified.

**Questions:**

Exploring advanced generative models or fine-tuning CycleGAN parameters could enhance the quality of synthetic H&E images, reducing potential artifacts and improving overall model performance. Incorporating validation steps to assess the fidelity of synthetic images compared to real H&E images would strengthen confidence in their utility.

Testing SHAP-CAT on additional cancer types and datasets would demonstrate its versatility and robustness.

---

### Official Review · Reviewer_vcC5 · 2024-11-05

**Soundness:** 3
**Presentation:** 2
**Contribution:** 2
**Rating:** 5
**Confidence:** 4

**Summary:**

This paper proposes a novel multimodal framework called SHAP-CAT, which utilizes a Shapley-value-based dimensionality reduction technique. Additionally, the authors employ virtual staining methods to generate a new modality, thereby enhancing the performance of the multimodal framework. The framework's effectiveness was validated on three public datasets.

**Strengths:**

1.The proposed Shapley-value-based dimensionality reduction technique and the multimodal framework for different WSIs demonstrate a certain degree of innovation.
2.The proposed method has been validated on multiple datasets.
3.Most of the writing and the method diagram are well done.

**Weaknesses:**

1.It appears that SHAP value is an approximation of Shapley value, proposed by (Lundberg & Lee, 2017)? What is the purpose of including Sections 4.2 and 4.3? Additionally, the writing of these two sections seems a bit confusing.
2. This paper claims in the title, abstract, and introduction that SHAP-CAT is an interpretable framework; however, it seems that SHAP is only used for feature dimensionality reduction. The reduced features remain abstract and uninterpretable. Additionally, this paper does not present any results or discussions regarding interpretability.
3. Are the top 32 dimensions of HE and rec_HE the same?
4. How were the results in Table 1 calculated? which dataset and modalities were used?
5. It seems that the BCI dataset only contains patch images of size 1024x1024, and the naming does not indicate which WSI the patches belong to. How were the results for methods like ABMIL calculated in Table 2? Do Tables 3 and 4 regarding the MIL results for the IHC4BC dataset also have similar issues?
6. What’s the difference between the BCI and IHC4BC datasets? Why is the experimental setup for BCI and IHC4BC different in this paper? For example, Table 3 lacks results for models like InceptionV3 and ResNet, while Table 2 includes them.
7. What are the implementation details of the pretrained CycleGAN? Why is the generated rec_IHC not considered as a modality to be included in the method?

**Questions:**

refer to the weaknesses.

---

### Official Review · Reviewer_H7L8 · 2024-11-05

**Soundness:** 1
**Presentation:** 3
**Contribution:** 1
**Rating:** 3
**Confidence:** 4

**Summary:**

This paper attempts to address multiple problems in multimodal learning on H&E and IHC whole slide images in computational pathology, namely limited paired datasets for multimodal learning, lack of interpretability, and lack of structural information in molecular features. To address this, the paper proposes a value imputation and augmentation methods (”virtual staining”) for different modalities and SHAP-based dimensionality reduction method to improve dimensionality after fusion using the Kronecker product.

**Strengths:**

- The authors identified two very important challenges in multimodal learning for whole slide images in lack of paired data and low interpretability of fusion methods.
- The paper is well-written and the notation is very clear.
- The methods on data imputation and augmentation are promising.

**Weaknesses:**

- How does the method help with explainability? You select 32 dimensions on an already aggregated slide-level representation. From a clinical perspective it would make sense to use SHAP to select the relevant patches/regions of interest, not dimensions within the embedding vector. You claim that the paper improves explainability but do not demonstrate any XAI capabilities of the model.
- The experimental setup seems flawed - you are comparing unimodal baselines with a multimodal model and don’t run the baselines with both modalities (which you could easily do with ABMIL and TransMIL). The manuscript also does not report standard deviations across folds which makes the statistical significance of the results impossible to judge.
- I disagree with the second identified challenge - this is not an open problem, as there are many multi-view learning models and most multiple-instance learning (MIL) pipelines can easily handle multiple slides or views. Additionally, it’s not strictly true that molecular features don’t account for structure (L49). For example, single-cell and spatial sequencing technologies do. Moreover, on the point of dismissing structure, the suggested approach also uses bag-level representations of a small subset of the image which is equally dismissive of the underlying tissue structure.
- The abstract could be clearer on which problems you are concretely addressing - you mostly talk about what methods you use but not why you use them.
- L92: “curve of dimensions” - do you mean curse of dimensionality?
- L98: you enlist IHC4BC-ER and IHC3BC-PR as separate datasets, but they are just different labels on the same dataset.
- On page 5 and 6, I don’t think it’s necessary to recite all properties of shap values from the original paper, as this provides little new information.

**Questions:**

- L349: You state that using 512-sized embeddings would generate 512^3-sized embeddings, but this is only true if you use the Kronecker product. There are many other fusion methods (e.g., low-rank tensor fusion) which do not suffer this problem. The most naive fusion method would be just to concatenate the vectors - have you tried these simpler approaches?
- How does this approach scale? SHAP pooling seems computationally expensive (you are fitting a separate model and then calculating the shap values).
- Have you compared wall time across the methods? My expectation is the SHAP-CAT takes much longer to run given the SHAP value calculation. Is a 0.003 AUC uplift compared to TransMIL worth the extract runtime?
- You mention that you use 5-fold Monte Carlo cross-validation but don’t report the standard deviation across folds. Would it be possible to report these?

---

### Official Review · Reviewer_8Yd3 · 2024-11-06

**Soundness:** 1
**Presentation:** 2
**Contribution:** 1
**Rating:** 3
**Confidence:** 5

**Summary:**

This paper introduces a novel methodology incorporating Shapley values and virtual staining techniques for a more holistic and interpretable whole-slide image (WSI) classification. The proposed approach demonstrates competitive performance, with ablation studies supporting the efficacy of individual components.

**Strengths:**

The integration of Shapley values for developing an interpretable WSI classification system is an interesting approach.

**Weaknesses:**

1. Technical writing requires improvement, particularly in terms of acronym definitions (e.g., WSI, IHC) and grammatical accuracy. The manuscript's writing quality falls below ICLR standards, with, but not limited to, several instances of imprecise language that I found in the paper: 1) Lines 39-40: Lack of specific detail regarding what kind of cross-modal information is missing in one modality and how the other modality can supplement, 2) Lines 47-48: Imprecise technical language, 3) Lines 51-52: Terminology should be "fusion techniques", 4) Line 92: Could be "curse of dimensionality", and 5) Line 199-200: Inconsistent capitalization and formatting issues

2. Insufficient justification for the proposed architectural choices (Why Shapley value is a good choice) and methodological decisions.

3. Statistical robustness is compromised by the absence of standard deviation metrics in experimental results, suggesting possible lack of repeated trials. The authors are suggested to provide at least one of standard deviations or significance test results.

4. Disproportionate space allocation to background material (e.g., cycleGAN, Shapley value) at the expense of methodology discussion and motivation.

5. Experimental evaluation concerns: Comparison against outdated baseline methods rather than contemporary state-of-the-art multimodal fusion approaches in WSI analysis, e.g., MCAT, MOTCAT, CMTA, MoME, SurvPath. Table 1: Unclear nomenclature (rand1, rand2, rand3) and inappropriate caption content. Table formatting issues (Table 3 out of the margin of the paper, Table 5 missing bold formatting for best results)

**Questions:**

1. How can the authors justify their claim that an additional modality provides complementary information, when the third modality is simply a reconstruction of the first modality (H&E slides)?

2. Could the authors provide concrete examples that demonstrate the model's interpretability capabilities?

---

### Note · Authors · 2024-11-15

I have read and agree with the venue's withdrawal policy on behalf of myself and my co-authors.